# Investigation of time to first presentation and extrahospital factors in the treatment of neovascular age-related macular degeneration: a retrospective cross-sectional study

Peng Yong Sim,[1] Sonul Gajree,[2] Baljean Dhillon,[1,3] Shyamanga Borooah[4,5]

[1]College of Medicine & Veterinary Medicine, The University of Edinburgh, Edinburgh, UK
[2]Ophthalmology Department, Gartnavel General Hospital, Glasgow, UK
[3]The Princess Alexandra Eye Pavilion, Edinburgh, UK
[4]Moorfields Eye Hospital, London, UK
[5]Centre for Clinical Brain Sciences, The University of Edinburgh, Edinburgh, UK

**Correspondence to**
Dr Peng Yong Sim;
pengyong91@gmail.com

## ABSTRACT

**Objectives** To assess the time from symptom onset to treatment for neovascular age-related macular degeneration (nvAMD) and to measure the awareness of AMD in Southeast Scotland.

**Design** Retrospective cross-sectional study.

**Setting** Secondary care, Southeast of Scotland.

**Methods** Patients treated with intravitreal therapy (IVT) for nvAMD in Southeast Scotland between 2013 and 2015 were identified using a treatment register. Notes were retrospectively reviewed. We measured time from: (A) symptom onset to first presentation at primary care, (B) referral to ophthalmic clinic appointment and (C) ophthalmic clinic appointment to first IVT treatment. To investigate AMD awareness, we performed a cluster random sample survey of patients visiting non-AMD ophthalmic clinics using a previously validated 12-item questionnaire.

**Results** 195 patients (mean age 78 years) were included in the study. The mean delays between the different stages—A, B and C—were 54.2 (95% CI ±13), 28.2 (95% CI ±4.0) and 31.5 (95% CI ±3.6) days, respectively. There was an additional mean delay of 7.5 (95% CI ±1.6) days when patients were indirectly referred by optometrists via general practitioners (P<0.05). 140 patients (mean age 78) participated in the awareness survey; 62.1% reported being 'aware' of AMD but only 37.3% described AMD symptoms correctly.

**Conclusions** There was a significant delay at every step of the nvAMD care pathway. The causes for this were multifactorial and included delays in first presentation to a healthcare provider, referral from primary care and initiation of secondary care treatment. Our data are likely to underestimate prehospital delays as a large number of cases are likely to have undefined symptoms and onset. We also identified suboptimal awareness of AMD which could account for a substantial delay in presentation from symptom onset. These findings highlight the need to address AMD awareness and the need for urgent treatment to prevent avoidable vision loss resulting from nvAMD.

## INTRODUCTION

Age-related macular degeneration (AMD) is the leading cause of vision loss in the developed world.[1] There are two main forms of AMD. The first is non-neovascular (dry) AMD

**Strengths and limitations of this study**

► Case notes of consecutive patients identified systematically using a treatment clinic register over 2 years.
► Demographic factors such as age, gender, education, social class and smoking status were taken into account for analyses of age-related macular degeneration (AMD) awareness.
► Unable to ascertain direct association between low disease awareness and delay in treatment for patients with neovascular AMD as different cohort of patients were examined.
► Due to retrospective analysis of notes from a disease register in this study, the number of patients who could be included was limited by the quality of note-keeping.

which accounts for the majority of AMD cases and results from the deposition of drusen (small yellow or white deposits) underneath the retina that eventually leads to the slow degeneration of retinal cells resulting in blindness. Neovascular (wet) AMD (nvAMD) accounts for the remaining cases of AMD and results from the development of new blood vessels deep to the retina which leak or bleed resulting in symptoms of new distortion or vision loss. nvAMD results in irreversible blindness if left untreated and accounts for 90% of the cases of blind registration resulting from AMD.[2] The main risk factors associated with AMD are age and smoking.[3] Cases of blindness resulting from AMD are predicted to increase together with an ageing population.[4]

An effective treatment is currently available to preserve vision in nvAMD in the form of intravitreal therapy (IVT) with antivascular endothelial growth factor agents.[5–8] These drugs have been shown to be effective in

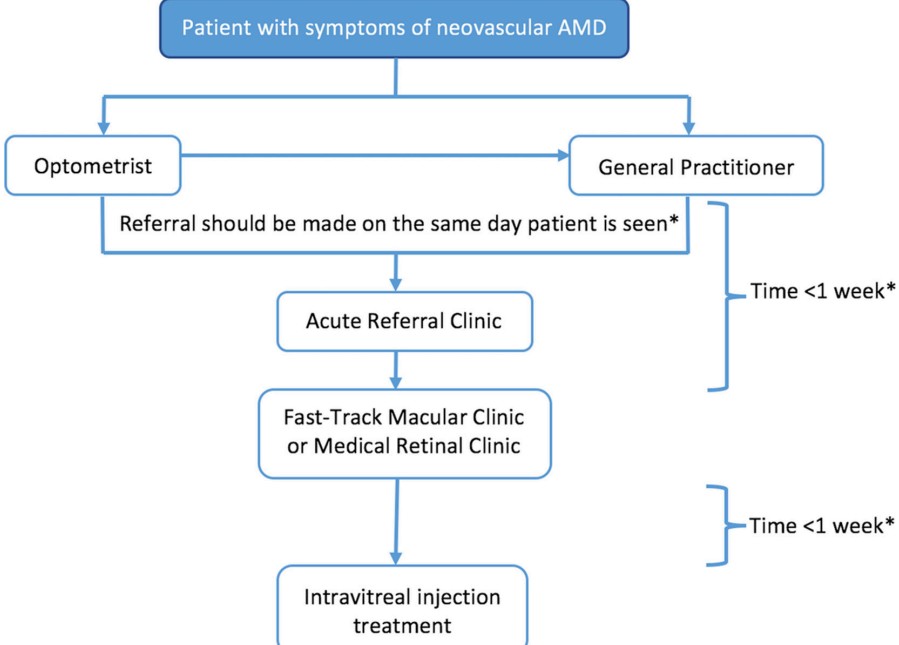

**Figure 1** Flow chart depicting the typical care pathway of a patient with neovascular age-related macular degeneration (AMD) in Southeast Scotland. *Based on the recommendations by the Royal College of Ophthalmologists in its 2013 AMD guideline.[14]

maintaining long-term vision in the majority of patients affected by nvAMD.[9] Delay in instituting IVT treatment in new cases of nvAMD has been shown to be one of the most important factors negatively impacting final visual outcome.[10 11] Consequently, the early diagnosis and treatment is crucial to improving visual outcomes in AMD and to reduce the social and economic burden of blindness resulting from the disease.[12 13]

Delays from symptom onset to treatment can be experienced at different stages of the patient care pathway for new onset nvAMD. These include: (1) time of first symptom onset to presentation at primary care practitioner, (2) time from primary care referral to presentation at ophthalmic clinic and (3) time from ophthalmic clinic to first IVT treatment (figure 1).[14] These early stages of the care pathway also represent the periods during which lesions may be most active and amenable to the benefits of therapy.[15]

There have been many published reports investigating intrahospital factors such as the time from first ophthalmic clinic visit to first IVT treatment.[10 11 16] However, there is a scarcity of literature reporting the extrahospital factors such as the time from symptom onset to presentation at ophthalmic clinic. In addition, despite its significance in causing blindness, limited research has been performed to investigate AMD awareness. An exploration of patient's awareness and knowledge of disease has been demonstrated in other chronic diseases such as stroke and cancer,[17 18] with increased awareness associated with improved patient outcomes.[19 20]

The primary objectives of this study were twofold: first, to assess the time between the different stages of the nvAMD care pathway in patients treated in Southeast

Scotland and second, to evaluate patients' awareness of AMD, its risk factors and treatment options.

## METHODS

Case notes of consecutive patients diagnosed and treated with IVT for nvAMD in NHS Lothian since September 2013 were identified using a treatment clinic register. A 2013 cut-off point was chosen to reflect the updated guidelines on AMD by the Royal College of Ophthalmologists (RCOphth) which were published at the time.[14] The guidelines recommended that all patients with suspected AMD should be seen by a retinal specialist within 1 week of referral and that treatment should commence within 1 week of first ophthalmic appointment (figure 1).

In this study, the main outcome measures were: (1) time from symptom onset to first presentation at primary care (ie, duration of visual symptoms before initial presentation), (2) time from primary care referral to ophthalmic clinic appointment and (3) time from ophthalmic clinic appointment to first IVT treatment. The main exclusion criteria were case notes with incomplete data and the coexistence of ocular comorbidities that gave rise to choroidal neovascularisation. This study was approved as part of a wider service evaluation which was accepted following review by the NHS Lothian Ophthalmology Quality Improvement Team on 8 October 2015.

In order to investigate patients' awareness of AMD, a cluster random sample of patients visiting ophthalmic clinics for non-AMD disease in NHS Lothian was surveyed using a 12-item questionnaire (see online supplementary file 1). The sample size required for the study was calculated using a power calculation (see online supplementary

file 2). Questions were adapted from a previously validated questionnaire[21] and served to ascertain each patient's knowledge of AMD and its risk factors. Patients were asked for their demographic details, including age, sex, education and postal code of residence. Socioeconomic deprivation scores (social class) were calculated for all patients from postal code data at the time of interview using the Scottish Index of Multiple Deprivation (SIMD).[22] The SIMD combines weighted data on seven domains (income, employment, education, housing, health, crime and geographical access) and is officially sanctioned by the Scottish Government as a measure of multiple deprivation.[23 24]

The first part of the questionnaire explored patients' familiarity with AMD and its risk factors. The second part enquired about patients' smoking status and their awareness of available treatments for AMD. Surveys were distributed and collected by the same researcher, who remained nearby to answer any questions about instructions. No additional assistance was provided. The survey was performed from 18 November 2015 to 31 November 2015 and data were analysed using Pearson $\chi^2$ tests except for education and social class where $\chi^2$ tests for trend were performed. Data analysis was done using IBM SPSS Statistics for Windows, V.23 (IBM, Armonk, New York, USA).

## RESULTS
### Delay in presentation, referral and treatment of AMD in Southeast Scotland
A total of 315 case notes were identified; 120 of the 315 were excluded after application of the exclusion criteria (see online supplementary data for the demographics and breakdown of excluded cases), leaving 195 case notes for analysis. One hundred and twenty (61.5%) patients were women, with a mean age of 78 years. Nearly all patients (187; 95.9%) presented with nvAMD affecting the first eye. The overall mean time from symptom onset to presentation was 54.2 (95% CI ±13) days. As for referrals to ophthalmology, 118 (60.5%) of these were direct from optometrists, 5 (2.6%) were direct from general practitioners (GPs) and 52 (26.7%) were made by optometrists via GPs. The remaining referrals were from other hospitals, other ophthalmology clinics and screening programmes.

The mean time from referral to ophthalmic clinic appointment was 28.2 days (95% CI ±4.0 days). There was a significant additional mean delay of 7.5 (P<0.05) (95% CI ±1.6) days when patients were referred from their optometrist via their GP. During clinic appointments, fundus fluorescein angiogram was performed in approximately one-third of patients (66/195). The mean time from clinic to first IVT treatment was 31.5 (95% CI ±3.6) days (figure 2).

### Awareness of AMD and its risk factors
The delay from symptom onset to first injection resulted from both intrahospital and extrahospital factors. We have already identified that when optometrists referred via the GPs instead of directly to the hospital eye service this resulted in a significant increased delay. However, even this delay is overshadowed by the mean delay from symptom onset to presentation at primary care service. In order to better understand patient factors that may have resulted in this delay in presentation we performed a questionnaire survey on patients with unrelated disease

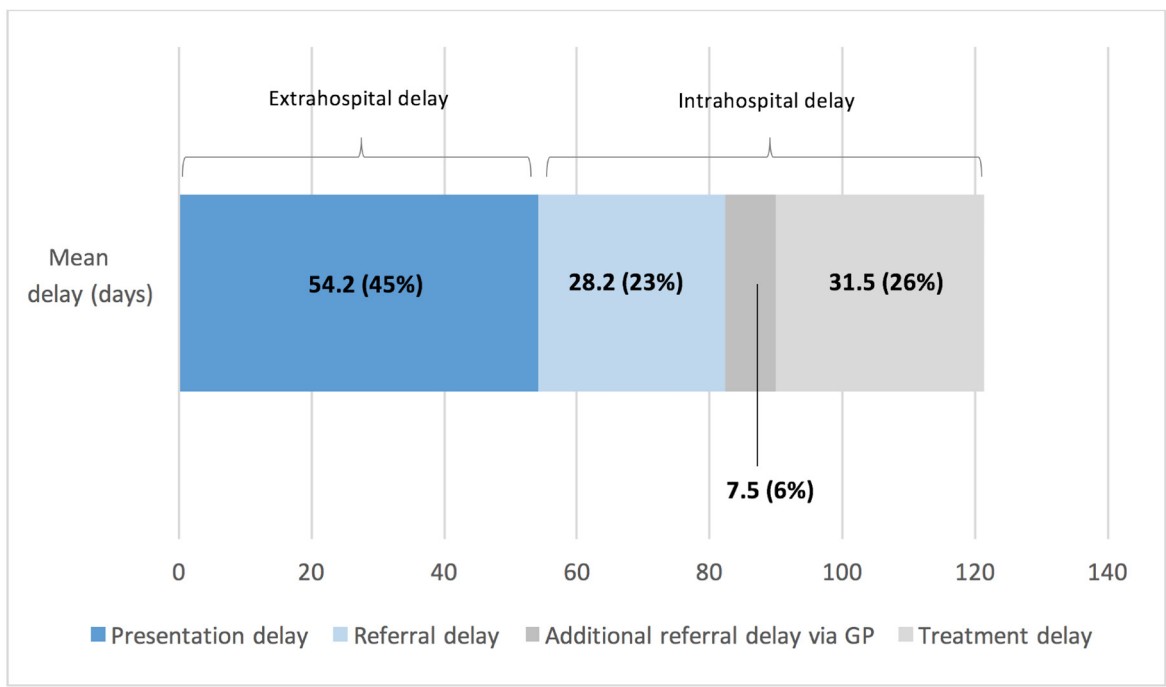

**Figure 2** Breakdown of the total delay (121.4 days) from symptom onset to treatment for patients with new neovascular age-related macular degeneration in Southeast Scotland. GP, general practitioner.

**Table 1** Demographic data of patients (N=140)

| Variable | n | % |
|---|---|---|
| Gender | | |
| Male | 61 | 43.6 |
| Female | 79 | 56.4 |
| Age (years) | | |
| <50 | 19 | 13.6 |
| ≥50 | 121 | 86.4 |
| Highest education level attained | | |
| Primary school | 4 | 2.9 |
| Secondary school | 78 | 55.7 |
| College | 31 | 22.1 |
| University degree | 27 | 19.3 |
| Social class | | |
| I | 13 | 9.3 |
| II | 25 | 17.8 |
| III | 20 | 14.3 |
| IV | 27 | 19.3 |
| V | 55 | 39.3 |
| Smoking status | | |
| Current smoker | 11 | 7.9 |
| Ex-smoker or non-smoker | 129 | 92.1 |

**Table 2** Respondents indicating awareness of AMD (N=140)

| Characteristic* | No of respondents indicating awareness/ total no (%) |
|---|---|
| Gender distribution | |
| Male | 31/61 (50.8) |
| Female | 56/79 (70.9) |
| P value | 0.015 |
| Age (years) | |
| <50 | 8/19 (42.1) |
| ≥50 | 79/121 (65.3) |
| P value | 0.053 |
| Highest education level attained | |
| Primary school | 4/7 (57.1) |
| Secondary school | 37/75 (49.3) |
| College | 23/31 (74.2) |
| University degree | 23/27 (85.2) |
| P value† | 0.001 |
| Social class | |
| I | 8/13 (61.5) |
| II | 18/25 (72.0) |
| III | 9/20 (45.0) |
| IV | 21/27 (77.8) |
| V | 31/55 (56.4) |
| P value† | 0.537 |
| Smoking status | |
| Current smoker | 6/11 (54.5) |
| Ex-smoker or non-smoker | 81/129 (62.8) |
| P value | 0.588 |

*Unless otherwise indicated, P values are derived using the Pearson $\chi^2$ test.
†Derived using the $\chi^2$ for trend.

in the eye service. A total of 142 patients were approached in non-AMD ophthalmic clinics. These clinics included glaucoma, ocular motility and general outpatient clinics. One hundred and forty patients agreed to participate. Two refused because of unwillingness and inability to understand the purpose of the questionnaire due to deafness respectively.

The cohort included 61 (43.6%) men and 79 (56.4%) women with a median age of 73 (range: 17–93) years, comprising all social classes. The education level of patients ranged from primary education to university degree. Details of the demographic data are given in table 1.

Of the 140 respondents, 87 (62.1%) reported being 'aware' of AMD. Fourteen (10%) had previously been diagnosed with AMD. Out of these 14 patients (71.4%), 10 were able to provide a correct description of the symptoms of AMD. For those patients without a prior diagnosis, only 47/126 (37.3%) were able to correctly report the symptoms of AMD. There was a significant difference when comparing the responses of those who had a previous diagnosis of AMD to those without AMD (P=0.013). Overall, female respondents were more likely than male respondents to report awareness of AMD (P=0.015) (table 2). Increased awareness of AMD was also seen with higher levels of education (P=0.001).

The top risk factor for AMD correctly considered by patients was age (127/140–90.7%). The other risk factors identified included smoking in 82 (58.6%), unprotected

UV exposure in 62 (44.3%), genetic predisposition in 62 (44.3%), vitamin deficiency in 54 (38.6%) and gender in 15 (10.7%).

Eighty-seven (62.1%) of patients thought that AMD was a treatable condition. However, only 20/87 (23%) were able to provide correct information on the available treatments (ie, eye injections and laser therapy). The majority of patients (91/140, 65%) considered opticians to be their first port of call if they had vision problems. Other healthcare professionals cited as first port of call included GPs in 28 (20%) and ophthalmologists in 21 (15%).

## DISCUSSION

The RCOphth has recently updated its guidance on suggested waiting times for IVT treatment in nvAMD in the hospital setting. It recommends that all patients

should be seen by a retinal specialist within 1 week of primary care referral, and should begin treatment within 1 week following this.[14] The new guidelines place increased importance on correct diagnosis and urgent referral from primary care and place increasing emphasis on hospital eye services to provide capacity for urgent new AMD cases in addition to the treatment of existing patients with nvAMD. However, this study finds that there are significant delays at each step of the nvAMD care pathway in south-east Scotland; both the waiting times from: (1) primary care referral to ophthalmic clinic and (2) initial ophthalmic assessment to treatment are about four times as long as the recommended gold standard. In addition, there is a further 1-week delay on average when indirect referrals are made by optometrists via GP. Similar findings have also been reported in previous studies which have demonstrated similar, if not longer, delays for intrahospital pathways (ie, from initial ophthalmic assessment to treatment).[10 11 16]

Delays from intrahospital pathways may be attributed to the inherent diagnostic and referral pathways within different healthcare systems. In Southeast Scotland, a new IT scheme linking community optometrists and eye clinics within hospitals across all of Scotland was introduced in 2010 following a successful pilot scheme in NHS Fife which allowed optometrists to make direct electronic referrals to ophthalmologists.[25] However, the system has yet to be fully integrated into all units. Our study has highlighted that there is still much room for improvement for both the primary care referral system, and also within the acute referral clinics themselves. The current electronic system still relies on a manual, ad hoc system for making referrals. An important step forward would be to develop a semiautomated referral system so that eye care providers can track patient referrals, obtain data on patient leakages and receive automatic notifications when there is lack of follow-up.

To our knowledge, this is the first study to evaluate the time from symptom onset to presentation at clinic (extrahospital pathway) for patients with nvAMD in the UK. There are, however, several limitations to the study. First, assessment of presentation delay might be difficult due to the retrospective nature of evaluation of symptom onset by patients. Second, the perception of symptoms is also highly subjective, often depending on factors such as existing cognitive function, ocular dominance of the affected eye and baseline visual acuity of the unaffected eye. Nevertheless, it is noteworthy that this time interval often varies widely between patients and is prolonged in most cases. Therefore, although less accurate than formal angiographic diagnosis, we thought it is important to investigate this time interval as it would be accessible to intervention.

Our findings demonstrate that presentation delay represents a major source of delay and accounts for the greatest proportion of the delay in the nvAMD care pathway in Southeast Scotland. This represents an important target for improvement to reduce vision loss resulting from delay in the nvAMD care pathway.[26] This delay is likely to be complex and multifactorial, involving patients, eye care providers and healthcare systems. Barriers to early presentation might include a lack of awareness of AMD among patients, self-examination by patients and screening of the disease by non-retina specialists. This can be further compounded by issues such as transport difficulties, age-related infirmity and a mismatch between patient expectations on speed of referral and recommended guidelines.[27] Further studies are warranted into the reasons underlying our findings in both primary care and hospital eye service environments in order that appropriate measures are taken to identify patients early and build service capacity accordingly.

At present, the diagnosis of new nvAMD, especially for the first affected eye, still very much relies on self-recognition of visual symptoms by patient themselves. This is, however, problematic as those affected in only one eye tend not to be aware of the visual change and may therefore remain 'asymptomatic' for a considerable length of time.[28] Indeed, this seemed to be case in our study in which nearly all patients presented with nvAMD affecting the first eye.

There is evidence to show that the best-corrected visual acuity at the time of diagnosis of nvAMD is worse for the first affected eye when compared with that of the second eye.[29] In addition, previous studies have shown that the visual prognosis of the first affected eye following 1 year of treatment is usually worse compared with that of the second affected eye in nvAMD.[30 31] These better outcomes of the second affected eye are most likely due to increased awareness and more frequent monitoring of the second eye as part of a systematic bilateral follow-up examination for the first affected eye. These factors would seemingly translate into a shorter delay in presentation for the second affected eye, but it should be noted that this association was not explored in our study and remains to be investigated. Nonetheless, the considerable delay in presentation for the first affected eye demonstrated in our study highlights the importance of early detection and treatment.

From the patient's perspective, the delay in symptom recognition can be addressed to a certain extent by self-examination. Patients, especially those with an increased risk of developing nvAMD, should be educated and made aware of symptoms such as new visual distortion and sudden reduction in vision. This can be achieved by encouraging patients to use suitable spaced self-tests of vision which examine one eye at a time to prevent compensation from the good eye. The standard Amsler test has long been recommended as the standard self-monitoring test, but there has been increasing reservation about its utility as a diagnostic tool due to its insufficient reliability and variable sensitivity.[32 33] The advent of more innovative, cost-saving technologies may circumvent these issues and make implementation of self-examination on a wider public scale more feasible in the near future.[34 35]

In this study, we chose to investigate patients' awareness of AMD because it is clear that a lack of disease awareness is a common factor for delayed presentation in other eye conditions such as glaucoma, retinal detachment and central retinal artery occlusion.[36 37] The only previous study to investigate AMD awareness in the UK population showed a low awareness (16%).[38] Our study adds to the existing literature by demonstrating that public awareness of AMD is still limited. Our survey shows that awareness of AMD is unacceptably low (37%), especially considering that this condition is the leading cause of blindness in developed countries.[1] The low awareness of AMD is also consistent with the low levels of awareness of AMD in other countries including Australia, Hong Kong, Singapore, Nepal, Bangladesh, China and the USA (range between 5% and 50.5%).[21 38–44] It is likely that our findings underestimate the true scale of lack of awareness among the general population because we sampled ophthalmic patients who, by virtue of being surveyed in an eye hospital, are presumably somewhat more attuned to common eye diseases. Our survey also highlights a low awareness of risk factors of AMD (other than age). However, this assessment could be limited by the lack of plausible distractors in the corresponding question which might have increased the respondent's chances of getting a correct answer(s), hence again underestimating the true scale of lack of awareness.

These findings are important given the severity of the consequences of delayed presentation in AMD and the ready availability of an effective treatment to prevent visual loss. We identified AMD-naive male patients and those with lower education levels to have a particularly low awareness of warning symptoms of AMD, suggesting the need for targeted intervention for these subgroups. As increased awareness can lead patients to seek appropriate medical care, improving awareness would logically lead to better visual prognoses for patients.

There is currently still a need for a unified national awareness campaign on AMD in the UK. A recent report by the Royal National Institute of Blind People highlighted that most initiatives at improving AMD awareness in the UK still operate at a local level.[27] Even then, these efforts often comprised of educational talks targeted at existing patients, rather than raising public awareness. The need for a national campaign has also been recognised by the Macular Society which has made increasing AMD awareness one of the main objectives of its 5-year national strategy.[45]

Although some progress has been made since,[46] there is still room for improvement. Current awareness interventions need to be further optimised for a sustained impact. A promising step would be the adoption of the multilayered approach as adopted by other developed countries.[47] This approach saw the use of a campaign which included a diverse range of activities such as promoting education programmes for patients and primary care, running a national advertising campaign and providing free mobile screening. The end of this focused campaign saw a dramatic increase in AMD awareness and the number of the population requesting fundus examination for symptoms of AMD.[47] The implementation of a similar public health strategy in the UK may achieve similar desirable effects, but further research is needed to evaluate the effectiveness of this approach in the UK population. Another important gap highlighted by our study is the underappreciated link between smoking and AMD. This represents a potent novel health promotional tool and awareness could be increased by incorporating information in existing campaigns with other smoking-related diseases.[48 49]

Lack of awareness and knowledge of correct referral among non-ophthalmologists is also problematic. This may account for the delay in referral demonstrated in our study. A recent national survey revealed that 32% of GPs felt 'deskilled' in diagnosing common eye conditions.[50] The same survey also showed that 38% of GPs felt that eyes are the most difficult part of the body to diagnose. Achieving a better alignment of ophthalmic knowledge between healthcare organisations and professionals will help improve understanding and management of common ophthalmic disorders for those in the front line of eye care.

## CONCLUSION

There is significant delay at every step of the care pathway for patients with nvAMD in Southeast Scotland. We also show that awareness and knowledge of AMD are suboptimal. This lack of AMD awareness could account for the long presentation delay of AMD to primary care. This suggests that efforts to educate the public regarding AMD may lead to earlier presentation and hence improved visual outcomes in patients.

**Contributors** BD and SB were involved in conception and design of the study. PYS and SG were involved in data acquisition, analysis and interpretation. PYS was involved in first draft of manuscript. SG, SB and BD were involved in revising and critically appraising manuscript. PYS, SG, BD and SB were involved in final approval for publication. BD and SB are guarantors.

**Funding** SB was sponsored by the Wellcome Trust Scottish Translational Medicine and Therapeutics Initiative (STMTI) scheme (grant no: R42141) during the course of this research.

**Competing interests** None declared.

**Ethics approval** NHS Lothian Ophthalmology Quality Improvement Team.

**Provenance and peer review** Not commissioned; externally peer reviewed.

**Data sharing statement** No additional data are available.

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
