## [Reviewer comments · BMJ Open]

ARTICLE DETAILS

TITLE (PROVISIONAL)	Investigation of time to first presentation and extra-hospital factors in the treatment of neovascular age-related macular degeneration: a retrospective cross-sectional study
AUTHORS	Sim, Peng Yong; Gajree, Sonul; Dhillon, Baljean; Borooh, Shyamanga

VERSION 1 – REVIEW

REVIEWER	Focke Ziemssen Center for Ophthalmology, Eberhard-Karl university Tuebingen Germany I have a potential conflict of interest after receiving speaker fees and consultation honoraria by Allergan, Alimera, Bayer and Novartis. However, the question of treatment delay is not directly related to the selection of treatment.
REVIEW RETURNED	25-Jun-2017

GENERAL COMMENTS	The topic is important. However the authors should suggest solutions in a more precise model: - information campaigns vs. screening vs. direct emergency consultation vs. training of optometrists and GPs Although it is not realistic to change the NHS, it would be helpful to compare the delay with other health systems (e.g. U.S. > Auckland, Germany and France). When was the survey conducted (assessing the patients' awareness)? The spatial relationship to the retrospective analysis of the delay might be of interest. What is the reason of the large portion of first eyes? It is very likely that there should be also cases with other causes of impairment as well as late stages of nAMD in the fellow eye. There are comparison of the first and second eye available including other frequent eye diseases besides nAMD. Is the FA examination not standard of care? Why did 129/165 did not receive any sufficient baseline diagnostic evaluation? The cohort of the survey is quite small. Did the authors make a sample size calculation in order to evaluate whether the small group is representative or not? Within the questionnaire, the order of 5. and 6. are not so wisely chosen. It would have been possible to ask first about the knowledge and then whether the participant might be affected. More distractors should have been used in question 8, in order to
--

	evaluate the precision of the knowledge. This should be critically discussed. There is no question on the family history or the association of the family history. In addition, the authors should have asked, whether the patients considered the disease to be an emergency. This is much more important than to assess the potential influential factors. I miss also any information regarding the self-examination (e.g. covering of one eye) within the survey. The terminology of 'wet' and 'dry AMD' is not well chosen (introduction). Please refer to more recent classifications and terms as covered in the literature. The authors should also comment shortly on the reasons for the intra-hospital delay. Please comment on the issue of transport and the patients' expectations vs. demands facing a reduced life expectancy.
--	---

REVIEWER	Hidenori Takahashi Associate Professor Department of Ophthalmology Jichi Medical University Japan
REVIEW RETURNED	05-Jul-2017

GENERAL COMMENTS	Peng Yong Sim, et al reported that time to first presentation and extra-hospital factors contribute to delays in the treatment of wet AMD. This study is important, because the treatment delay affects visual prognosis. To improve this manuscript, the reviewer suggests one thing. Onset was based on the patients' report. Some patients do not notice onset due to AMD on non-dominant eye, slow progression of AMD, and dementia. At the reviewer's hospital, 5 to 10 % patients can not report onset day, despite ophthalmologists repeatedly ask when. In this manuscript, "120 of the 315 were excluded due to incomplete data and the co-existence of ocular comorbidities" is stated in the method section. No statement on results and discussion (limitation) section. Detail of excluded cases is important for study bias. It is necessary to state details of excluded cases, how many patients with incomplete data, a breakdown of incomplete data, etc.
---

REVIEWER	Steven Lane Liverpool University England No Competing Interest
REVIEW RETURNED	06-Jul-2017

GENERAL COMMENTS	Introduction Could authors explain what they mean by deep to the retina
--

VERSION 1 – AUTHOR RESPONSE

Reviewer: 1

Reviewer Name: Focke Ziemssen

Institution and Country: Center for Ophthalmology, Eberhard-Karl university Tuebingen, Germany

Competing Interests: I have a potential conflict of interest after receiving speaker fees and consultation honoraria by Allergan, Alimera, Bayer and Novartis.

However, the question of treatment delay is not directly related to the selection of treatment.

Please leave your comments for the authors below

1. The topic is important. However the authors should suggest solutions in a more precise model: - information campaigns vs. screening vs. direct emergency consultation vs. training of optometrists and GPs.

Response: We thank the reviewer for his helpful advice. In addition to the multi-layered approach proposed (education programmes, national advertising campaigns and free mobile screening), we have added a new paragraph to highlight the need to improve ophthalmic knowledge amongst optometrists and GPs.

“Lack of awareness and knowledge of correct referral among non-ophthalmologists is also problematic. This may account for the delay in referral demonstrated in our study. A recent national survey revealed that 32% of GPs felt “de-skilled” in diagnosing common eye conditions. The same survey also showed that 38% of GPs felt that eyes are the most difficult part of the body to diagnose. Achieving a better alignment of ophthalmic knowledge between healthcare organisations and professionals will help improve understanding and management of common ophthalmic disorders for those in the front line of eye care.”

2. Although it is not realistic to change the NHS, it would be helpful to compare the delay with other health systems (e.g. U.S. > Auckland, Germany and France).

Response: The comparison of delays for intra-hospital pathways (i.e. from initial ophthalmic assessment to treatment) with other health systems has been mentioned in the last sentence of the first paragraph under the discussion section. Reference has been made to three studies from Australia, Japan and Spain which showed average delays of 10-12 weeks (77 days), 92 days and 2.3 months (70 days) respectively. To the best of our knowledge, we are not aware of any other published study in other countries investigating delay within the AMD care pathway.

3. When was the survey conducted (assessing the patients' awareness)? The spatial relationship to the retrospective analysis of the delay might be of interest.

Response: The last paragraph under the methods section states the dates the survey was conducted (November 18 to 31, 2015). We appreciate that the current sentence might be misinterpreted so we have made it clearer - "The survey was performed from 18th November 2015 to 31st November 2015." Retrospective analysis of both delay and survey data were performed in January 2016.

4. What is the reason of the large portion of first eyes? It is very likely that there should be also cases with other causes of impairment as well as late stages of nAMD in the fellow eye. There are comparison of the first and second eye available including other frequent eye diseases besides nAMD.

Response: The reason for the large portion of patients presenting with first eyes in our study is because we only included patients with information on presentation delay (i.e. symptom onset to first presentation at primary care). This was the case for patients presenting with first eyes as they were able to report an estimate of symptom duration. On the other hand, most of those with second eye involvement were diagnosed incidentally on routine scheduled follow-up following first eye diagnosis which meant that there was no available information on presentation delay. As for the presence of other frequent eye diseases, we have aimed to minimise the influence of this by excluding patients who had co-existing ocular comorbidities that gave rise to choroidal neovascularisation (e.g. diabetic maculopathy, retinal vein occlusions). This has been mentioned in the methods section. However, we did not take into account the presence of other common eye diseases (e.g. cataract, glaucoma) into our analysis and hence, could not accurately ascertained the impact of these co-existing conditions on presentation delay.

5. Is the FA examination not standard of care? Why did 129/165 did not receive any sufficient baseline diagnostic evaluation?

Response: The use of fluorescein angiograph (FA) was not part of the routine neovascular AMD care pathway in south-east Scotland during the study period. This investigation was only used when there was any degree of diagnostic doubt. Otherwise, the clinical evaluation and optical coherence tomography (OCT) findings consistent with neovascular AMD was the approach taken by the treating ophthalmologist in making the diagnosis.

6. The cohort of the survey is quite small. Did the authors make a sample size calculation in order to evaluate whether the small group is representative or not?

Response: Based on a global survey investigating AMD awareness by AMD Alliance International, the level of awareness in the UK was 16% in 2005. Allowing for increase in awareness over time (demonstrated by studies in other countries), hence assuming a slightly higher level of awareness (~25%) in south-east Scotland, we would have a power of 80% to detect this with a total sample size of 118 patients.

7. Within the questionnaire, the order of 5. and 6. are not so wisely chosen. It would have been possible to ask first about the knowledge and then whether the participant might be affected. More distractors should have been used in question 8, in order to evaluate the precision of the knowledge. This should be critically discussed.

There is no question on the family history or the association of the family history. In addition, the authors should have asked, whether the patients considered the disease to be an emergency. This is much more important than to assess the potential influential factors. I miss also any information regarding the self-examination (e.g. covering of one eye) within the survey.

Response: The order of question 5 and 6 were adopted from a previously validated questionnaire. In retrospect, we agree that this order might not have followed a logical progression and that the incorporation of the additional questions suggested above (especially patient's perceived severity of AMD) would have been very useful and will take note of this advice for future studies.

Self-examination practice was not explored in this survey as we set out specifically to explore patient's awareness of AMD, its risk factors and treatment options. However, it would be useful to ascertain this information as it would have highlighted a potential area of improvement for awareness campaigns. We have now commented of the lack of distractors in exploring awareness of AMD risk factors in the text by adding:

“Our study also highlights a low awareness of risk factors of AMD (other than age). However, this assessment could be limited by the lack of plausible distractors in the corresponding question which might have increased the patient's chances of getting a correct answer(s), hence again underestimating the true scale of lack of awareness.”

8. The terminology of 'wet' and 'dry AMD' is not well chosen (introduction). Please refer to more recent classifications and terms as covered in the literature.

Response: We have now updated the terminology of 'dry' and 'wet' to 'non-neovascular' and 'neovascular' respectively to reflect the latest classification system.

9. The authors should also comment shortly on the reasons for the intra-hospital delay.

Response: We have added the following paragraph under the discussion section to address the reasons for intra-hospital delay:

“Delays from intra-hospital pathways may be attributed to the inherent diagnostic and referral pathways within different healthcare systems. In south-east Scotland, a new IT scheme linking community optometrists and eye clinics within hospitals across all of Scotland was introduced in 2010 following a successful pilot scheme in NHS Fife which allowed optometrists to make direct electronic referrals to ophthalmologists. However the system has yet to be fully integrated into all units. Our study has highlighted that there is still much room for improvement for both the primary care referral system, and also within the acute referral clinics themselves. The current electronic system still relies on a manual, ad-hoc system for making referrals. An important step forward would be to develop a semi-automated referral system so that eye care providers can track patient referrals, obtain data on patient leakages and receive automatic notifications when there is lack of follow-up.”

10. Please comment on the issue of transport and the patients' expectations vs. demands facing a reduced life expectancy.

Response: We have added a further sentence to the discussion to address the reviewers comment:

“This can be further compounded by issues such as transport difficulties, age-related infirmity and a mismatch between patient expectations on speed of referral and recommended guidelines.”

Reviewer: 2

Reviewer Name: Hidenori Takahashi

Institution and Country: Associate Professor, Department of Ophthalmology, Jichi Medical University, Japan

Competing Interests: None declared

Peng Yong Sim, et al reported that time to first presentation and extra-hospital factors contribute to delays in the treatment of wet AMD. This study is important, because the treatment delay affects visual prognosis.

1. To improve this manuscript, the reviewer suggests one thing. Onset was based on the patients' report. Some patients do not notice onset due to AMD on non-dominant eye, slow progression of AMD, and dementia. At the reviewer's hospital, 5 to 10 % patients can not report onset day, despite ophthalmologists repeatedly ask when.

Response: We have now acknowledged this limitation in the discussion:

“To our knowledge, this is the first study to evaluate the time from symptom onset to presentation at clinic (extra-hospital pathway) for patients with neovascular AMD in the UK. There are however several limitations to the study. Firstly, assessment of presentation delay might be difficult due to the retrospective nature of evaluation of symptom onset by patients. Second, the perception of symptoms is also highly subjective, often depending on factors such as existing cognitive function, ocular dominance of the affected eye and baseline visual acuity of the unaffected eye. Nevertheless, it is noteworthy that this time interval often varies widely between patients and is prolonged in most cases. Therefore, although less accurate than formal angiographic diagnosis, we thought it is important to investigate this time interval as it would be accessible to intervention.”

2. In this manuscript, "120 of the 315 were excluded due to incomplete data and the co-existence of ocular comorbidities" is stated in the method section. No statement on results and discussion (limitation) section.

Response: We have addressed the reviewer's comments by including this information in the supplementary data – “A total of 315 case notes were identified; 120 of the 315 were excluded after application of the exclusion criteria (see Supplementary Data for the demographics and breakdown of excluded cases), leaving 195 case notes for analysis.”

3. Detail of excluded cases is important for study bias. It is necessary to state details of excluded cases, how many patients with incomplete data, a breakdown of incomplete data, etc.

Response: Two tables have now been added under “Supplementary files” to compare demographic data for included vs excluded case notes and show breakdown of excluded case notes. In summary, the included patients and excluded patients had a similar demographic profile. Of the 120 case notes excluded from the study, 23 had co-existence of ocular comorbidities that gave rise to choroidal neovascularisation, 76 had missing information on symptom duration prior to presentation and 21 were lost to follow-up.

Reviewer: 3

Reviewer Name: Steven Lane

Institution and Country: Liverpool University, England

Competing Interests: None

1. Introduction

Could authors explain what they mean by deep to the retina

Response: This has now been made clearer in the introduction – “The first is non-neovascular (dry) AMD which accounts for the majority of AMD cases and results from the deposition of drusen (small yellow or white deposits) underneath the retina that eventually leads to the slow degeneration of retinal cells resulting in blindness.”

VERSION 2 – REVIEW

REVIEWER	Focke Ziemssen University Eye Hospital Tuebingen, Germany None (directly related to the topic, however there are some conflict of interest concerning the manufacturers of drugs)
REVIEW RETURNED	29-Aug-2017

GENERAL COMMENTS	The manuscript still suffers from some uncertainties:  - The authors should clarify that the analysis did only evaluate the tip of the iceberg. By excluding the large group with undefined or unreported onset of symptoms, the study cannot assess the extent of the problem. There is a 'telescopic' phenomenon that patients underestimate the duration of symptoms beside the many other limitations (first eye, limited knowledge of eye diseases, etc).  - For the interfaces, the problems cannot be clearly assigned. Is it an intra-hospital problem, if the appointments cannot be offered. What is the influence of the transmitted information (optician, GP)? - Was it really wise to use this environment for assessing the awareness? The most important question would be the knowledge about the urgency of the treatment (not only the facts that there are treatments and the existence of the symptoms). I would prefer, if those limitations would be more clearly integrated and emphasized in the abstract. It should be clarified that avoidable blindness and loss of vision is caused by this frequent and serious problem.
--

REVIEWER	Hidenori Takahashi Jichi Medical University, Japan
REVIEW RETURNED	13-Aug-2017

GENERAL COMMENTS	The manuscript has been satisfactory improved.
--

VERSION 2 – AUTHOR RESPONSE

Reviewer: 1

Reviewer Name: Focke Ziemssen

Institution and Country: University Eye Hospital Tuebingen, Germany

Competing Interests: None (directly related to the topic, however there are some conflict of interest concerning the manufacturers of drugs)

The manuscript still suffers from some uncertainties:

Comment: The authors should clarify that the analysis did only evaluate the tip of the iceberg. By excluding the large group with undefined or unreported onset of symptoms, the study cannot assess the extent of the problem. There is a 'telescopic' phenomenon that patients underestimate the duration of symptoms beside the many other limitations (first eye, limited knowledge of eye diseases, etc).

Response: We thank the reviewer for his helpful advice. We have now edited the conclusion in the abstract to address this comment and others below as requested:

“There was a significant delay at every step of the nvAMD patient pathway. The causes for this were multifactorial and included delays in first presentation to a healthcare provider, referral from primary care and initiation of secondary care treatment. Our data is likely to underestimate pre-hospital delays as a large number of cases are likely to have undefined symptoms and onset. We also identified suboptimal awareness of AMD which could account for a substantial delay in presentation from symptom onset. These findings highlight the need to address AMD awareness and the need for urgent treatment to prevent avoidable vision loss resulting from nvAMD.”

Comment: For the interfaces, the problems cannot be clearly assigned. Is it an intra-hospital problem, if the appointments cannot be offered. What is the influence of the transmitted information (optician, GP)?

Response: We would like to direct the reviewer to the fourth paragraph of the discussion which has already covered this topic. The delay to first presentation detected in our study is likely to be complex and multifactorial. The investigation of reasons underlying this delay was however beyond the scope of this study. We have updated the conclusion of the abstract to encompass this (see above). We agree with the reviewer that this an area needing further research and have added the following sentence under the discussion section to address this gap:

“Further studies are warranted into the reasons underlying our findings in both primary care and hospital eye service environments in order that appropriate measures are taken to identify patients early and build service capacity accordingly.”

Comment: Was it really wise to use this environment for assessing the awareness? The most important question would be the knowledge about the urgency of the treatment (not only the facts that there are treatments and the existence of the symptoms).

Response: We acknowledge that conducting the survey in the hospital setting is a possible limitation and have previously highlighted this in the discussion section: “It is likely that our findings underestimate the true scale of lack of awareness among the general population because we sampled ophthalmic patients who, by virtue of being surveyed in an eye hospital, are presumably somewhat more attuned to common eye diseases.”

As for patient's knowledge of the urgency of treatment, this was not explored in this survey as set out specifically to explore patient's awareness of AMD, its risk factors and treatment options. Nonetheless, we agree that the evaluation of this information is crucial and will take note of this advice in a future study.

Comment: I would prefer, if those limitations would be more clearly integrated and emphasized in the abstract. It should be clarified that avoidable blindness and loss of vision is caused by this frequent and serious problem.

Response: Limitations now highlighted in abstract (see response to 1st comment).

VERSION 3 – REVIEW

REVIEWER	Focke Ziemssen Department for Ophthalmology, Eberhard-Karl university Tuebingen The reviewer received speaker's fee and consultancy honoraria by some manufacturers of drugs of nvAMD: Bayer, Novartis.
REVIEW RETURNED	08-Oct-2017
GENERAL COMMENTS	Thank you for the revised versions. Although I would have addressed some of the limitations more in detail, I think that the actual version of the manuscript is newsworthy.